# Effect of High Dietary Level (8%) of Fish Oil on Long-Chain Polyunsaturated Fatty Acid n-3 Content in Pig Tissues and Plasma Biochemical Parameters

**DOI:** 10.3390/ani10091657

**Published:** 2020-09-15

**Authors:** Tomas Komprda, Miroslav Jůzl, Milena Matejovičová, Lenka Levá, Markéta Piechowiczová, Šárka Nedomová, Vendula Popelková, Pavla Vymazalová

**Affiliations:** 1Department of Food Technology, Mendel University in Brno, 613 00 Brno, Czech Republic; milena.matejovicova@mendelu.cz (M.M.); marketa.piechowiczova@mendelu.cz (M.P.); sarka.nedomova@mendelu.cz (Š.N.); xpopelk8@node.mendelu.cz (V.P.); xvymaza5@node.mendelu.cz (P.V.); 2Department of Immunology, Veterinary Research Institute, 621 00 Brno, Czech Republic; leva@vri.cz

**Keywords:** docosahexaenoic acid, plasma triacylglycerols, neutrophils, transaminases, Warner-Bratzler shear force

## Abstract

**Simple Summary:**

Fish oil is a source of “healthy” polyunsaturated fatty acids that decrease the risk of cardiovascular and other chronic diseases in humans. One possible way to increase consumption of these fatty acids is to incorporate them into pork via pig-feed. We enriched the pigs’ diet with 8% fish oil and found a several-times higher content of the above-mentioned fatty acids in the pig meat in comparison with controls. The meat of the fish oil-fed pigs was also more tender. We also used the same pigs as a model for testing the effects of the high level of dietary fish oil on selected markers of their state of health and detected an increased level of plasma lipids (risk factor for cardiovascular diseases) and a possible overreaction of the immune system. In conclusion, we can recommend consumption of polyunsaturated fatty acids-enriched “functional” pork, but direct human consumption of fish oil should be limited and in accordance with the recommendation of a moderate intake.

**Abstract:**

There were two objectives of the present study using dietary fish oil (FO) in pigs: to use pigs as a model for studying the effects of high FO doses on selected physiological markers; and to evaluate the physical traits and nutritive value of pork enriched with long-chain polyunsaturated fatty acids n-3. Two groups of six female pigs were fed for 30 days with either a standard feed mixture (control, C) or the same mixture supplemented with 8% FO (F). Physical characteristics of the muscle, fatty acid deposition in tissues and selected hematologic and plasma markers were tested. The daily weight gain of the F-pigs was lower in comparison with controls (*p* < 0.05). Dietary fish oil decreased Warner-Bratzler shear force of the longissimus muscle (*p* < 0.01). The eicosapentaenoic and docosahexaenoic acid content was higher (*p* < 0.05) in all tested F-tissues. Dietary fish oil had no effect on plasma cholesterol (*p* < 0.05), but it increased plasma triacylglycerol levels by 260% (*p* < 0.05), and increased counts of leukocytes, neutrophils, and eosinophils in the blood plasma (*p* < 0.05). In conclusion, high dietary FO improved the texture and nutritive value of meat, but negatively affected plasma biochemical parameters.

## 1. Introduction

The nutritional and physiological importance of long-chain polyunsaturated fatty acids n-3 (LC-PUFA n-3) in the prevention of chronic degenerative diseases in humans is generally recognized [1]. The following positive health effects are ascribed to LC-PUFA n-3: antithrombogenic, anti-inflammatory, hypotriglyceridemic, prevention of arrhythmias, and amelioration of autoimmune disorders, including rheumatoid arthritis [2]. The present-day Western diet is low in PUFA n-3, with a ratio PUFA n-6/n-3 of 15-20:1, instead of an optimal value of 1:1 [3]. A value as close as possible to 1:1 is considered protective against degenerative pathologies [4]. A daily intake of 250 mg EPA (eicosapentaenoic acid, 20:5 n-3) and DHA (docosahexaenoic acid, 22:6 n-3) is recommended in the European Union [5].

A natural source of LC-PUFA n-3 is fatty fish, the consumption of which is very low in many parts of the world. A feasible alternative to fatty fish consumption is to manipulate the fatty acid composition of meat and meat products towards a more favorable n-6/n-3 ratio [3,6]. In monogastric animals, tissue fatty acid profiles closely reflect the fatty acid profiles of their diets [7]. Changes in animal nutrition by addition of fat sources with a favorable ratio of PUFA n-6/n-3 can lead to positive changes in the composition of animal products, such as milk [8], eggs [9], and meat [10].

However, it is ineffective to enrich pork, probably the most-consumed meat worldwide, with LC-PUFA n-3 by a dietary supplementation with the sources of their precursor, indispensable α-linolenic acid (ALA), as the conversion rate of ALA to LC-PUFA n-3 is very limited in pigs [11]. According to these authors [11], a high intake of ALA barely affects DHA concentrations in the intramuscular fat of pork, so direct supplementation with EPA- and DHA-rich sources is required. ALA competes with 24:5 n-3 for the Δ6-desaturase. This competition decreases DHA synthesis, which can then be retroconverted into EPA; consequently, the deposition of de novo synthesized DHA in tissues is low. In contrast, feeding pigs with fish oil yields high proportions of EPA, DPA, and DHA [12]. Fish oil or fish meal can generally be used as a direct dietary source of EPA and DHA [7].

Furthermore, because the morphology and physiology of the organs of humans and pigs are similar [13], pigs are, in addition to being a possible source of LC-PUFA n-3 enriched products for human consumption, an excellent animal model for studying the effects of LC-PUFA n-3 on parameters correlating with the risk of chronic degenerative diseases in humans. For example, in our previous study [14], a porcine model was found to be superior to a rat model, as far as the effect of dietary fish oil on plasma lipid profiles is concerned.

It follows from the results of our own experiments using the in vivo models addressing markers of chronic degenerative diseases in humans that dietary fish oil supplemented at an usual level of 25 g/kg of the pig’s diet was neither able to improve plasma lipid markers in comparison with saturated palm oil [14] nor to sufficiently ameliorate inflammation [15].

Therefore, we intended in the present experiment to supplement the pigs’ diet with fish oil at an amount three to four times higher than the above-mentioned level of 25 g/kg, and we considered two aspects of the enrichment. On the one hand, pigs were used as a source of pork with not an ideal composition of fatty acids, which can be modified by a substantial dietary intervention. Simultaneously, pigs were used as an animal model for testing selected physiological markers pertinent to human health.

Consequently, the objective of the present experiment was to test two hypotheses: (1) the nutritive value of pork (content of LC-PUFA n-3; PUFA n-6/n-3 ratio) can be substantially increased by dietary intervention using sufficient amounts of fish oil, i.e., 80 g/kg of feed; (2) selected physiological markers, including plasma lipid levels, can be improved (using pigs as an animal model of human nutrition) by dietary intervention with higher than usual levels of fish oil.

## 2. Materials and Methods

### 2.1. Animals, Diets, Sample Collection

Twelve female pigs of the hybrid Large White (50%) × Landrace (50%; Bioprodukt Knapovec a.s., Ústí nad Orlicí, Czech Republic) at the age of 12 weeks with a mean live weight of 52.3 kg were used. The pigs were housed in an experimental stable in four floored indoor pens of 290 × 343 cm (the height of the room was 280 cm) containing three animals each. The experiment was conducted in compliance with the Czech National Council Act No. 246/1992 Coll. to protect animals from cruelty and with the Amended Act. No. 162/1993 Coll., and was approved by the Commission to Protect Animals against Cruelty of Mendel University in Brno and the Ministry of Agriculture of the Czech Republic under the statement No. 16252-MZE-17214.

The pigs were divided into two groups of six animals each, and the pigs within each group were further divided into two pens of three animals each. The control group (C) was fed a standard commercial feed mixture for pig fattening (De Heus, Marefy, Czech Republic), while the experimental group (F) was given the same feed mixture enriched with 8% (w/w) of fish oil (commercial *jecoris aselli oleum*; vitamin A: 1096 IU/g, vitamin D_3_: 224 IU/g). The composition of the diets is presented in Table 1, and the fatty acid content of the fish oil and the diets in Table 2. The animals had free access to drinking water and were fed twice daily ad libitum. By subtracting leftovers, the net feed consumption and intake was measured per pen (not individually). Fish oil was always freshly admixed into the feed mixture twice a day, separately for the morning and afternoon feeding. The pigs were weighed at weekly intervals, and the fattening lasted for 30 days.

Blood samples were drawn from the *vena cava cranialis* of each pig for the hematological analysis and plasma separation on the 30th day of the fattening, after fasting. The blood was collected into tubes with *heparinum natricum* (25 IU/mL of blood; Zentiva, Prague, Czech Republic). The samples were immediately centrifuged at 1300× *g* for 15 min at 4 °C, and the plasma samples were kept at −80 °C until the biochemical parameters were determined.

After that, the pigs were transported to the commercial abattoir, slaughtered, and the weight at delivery, carcass weight, and the weights of shoulder and leg, respectively, were measured. The parts were then transported to the university meat production lane and aliquot samples (300–500 g) of muscle (*musculus longissimus lumborum et thoracis*; collected at the last rib), liver, heart, and visceral (pelvic) adipose tissue (VAT) were taken.

### 2.2. Physical Traits

The physical traits of *m. longissimus* were evaluated. Dry matter was determined by oven drying at 105 °C to constant weight, while protein content (N × 6.25) was measured via the Kjeldahl method with the Kjeltec System 8200 (FOSS NIRSystems Inc., Hillerød, Denmark). The total lipid content was quantified as hexane/2-propanol extract, according to [14]. The pH_1_ values were measured (between the 10th and 11th rib) with a pH portable meter Knick PORTAVO 907 multi pH (Knick, Berlin, Germany) at the slaughterhouse one hour post mortem, and pH_24_ after transport to the university laboratory 24 h post mortem (the samples were first stored in a refrigerated transport box and subsequently in a refrigerator). Each muscle sample was measured by a puncture in three different parts. The apparatus was calibrated using the pH 7 and pH 4 buffers.

The 150 g aliquot sample after dissection was used for evaluation of the drip loss. The sample was weighed with an accuracy to 0.0001 g, placed into a polyethylene bag, and kept in a refrigerator for 24 h at 5 °C. After that, the sample was wiped using a filter paper and weighed again. The results were expressed as a percentage of the drip loss.

The color characteristics L* (lightness from black to white), a* (green to red), and b* (blue to yellow) were measured (after the appropriate bloom time: 15 min after the fresh cut) within the CIELAB color space [17] using a spectrophotometer Minolta CM-3500d (Konica Minolta, Osaka, Japan) equipped with the D65 illuminant, 8° observer angle, and a slit of 8 mm. Muscle samples were cut into 20 mm slices and the color characteristics were measured twice at three places in each sample.

The firmness of the muscle samples was quantified via the Warner-Bratzler shear force (WBS). The aliquot samples were cut into 1 cm × 1 cm × 5–6 cm pieces and cooked in a boiling water bath up to a final core temperature of 70 °C, before being cooled to room temperature and then cross-cut by the Warner-Bratzler shearing device (crosshead speed of 1 mm s^−1^) perpendicularly to the muscle fiber direction using the TIRATEST 27 O25 apparatus (TIRA Maschienenbau GmbH, Schalkau, Germany). The hardness of the samples of 1 cm × 1 cm × 1 cm was measured by compression using a 50% compression rate, with a crosshead speed of 50 mm using the same apparatus.

### 2.3. Fatty Acid Analysis

Two samples were taken from two different parts of each tissue of each pig (regarding muscle, it was the proximal and caudal ends of the *m. longissimus*, respectively). Total lipids were extracted from each of these two samples by the hexane/2-propanol mixture [14] and each extract was analyzed in duplicate. The analysis was performed based on the procedure outlined by [14] with the following exceptions: Fatty acid methyl esters (FAMEs) were separated using an Agilent 6890 gas chromatograph with an autosampler (Agilent Technologies, Wilmington, DE, USA) on the ZB FAST FAME capillary column (20 m × 0.18 mm × 0.15 µm; Phenomenex, Torrance, CA, USA). The injector and detector (flame ionization detector) temperatures were 250 °C and 260 °C, respectively. The temperature program was as follows: the start temperature was 80 °C, held for two minutes; 10 °C/min until 160 °C; 3 °C/min until 185 °C; 30 °C/min until 240 °C, held for two minutes. The flow rate of the carrier gas (N_2_) was 0.5 mL/min, and the split ratio was 1:100. The NU-CHECK 455—16 FAME (NU-CHECK Prep, Inc., Elysian, MN, USA) was used as an external standard for FAME identification.

### 2.4. Hematological Markers

Hematologic traits (hematocrit, hemoglobin, erythrocytes, leukocytes, lymphocytes, and monocytes, and neutrophil, eosinophil and basophil granulocytes) were measured using a BC-2800Vet hematological analyzer (Mindray, Bio-Medical Electronics, Shenzhen, China). A differential counting was performed microscopically, and white blood cells were counted under a microscope using the Pappenheim panoptic staining technique with the May—Grünwald and Romanowski—Giemsa stains.

### 2.5. Biochemical Determinations

The following biochemical traits were determined in plasma using a biochemical analyzer BS-200 (Mindray, Bio-Medical Electronics, Shenzhen, China): the activities of alanine transaminase, aspartate transaminase, and alkaline phosphatase, and the concentration of total plasma protein, albumin creatinine, urea, glucose, triacylglycerols (TAG), total cholesterol (TC), high-density lipoprotein cholesterol (HDLC), and low-density lipoprotein cholesterol (LDLC).

### 2.6. Statistical Analysis

The data were expressed as the mean ± standard error of the mean. Based on the normality of the data distribution (evaluated by Shapiro—Wilk test), the differences between the C- and F-groups were tested by either a Mann—Whitney U test or an independent-group t-test using the STATISTICA 12 package (StatSoft, Tulsa, OK, USA). The means were considered to be different at a significance level of *p* < 0.05.

## 3. Results

### 3.1. Growth Performance

Dietary fish oil did not significantly affect final live weight of the pigs (*p* > 0.05), but the daily weight gain (DWG) of the F-pigs was lower in comparison with the controls (*p* < 0.05). Moreover, dietary fish oil tended to decrease the weight at delivery, yield, and leg weight (Table 3).

### 3.2. Physical Characteristics of the Muscle

The *musculus longissimus lumborum et thoracis* was examined. The dry matter (26.86 and 26.38%), fat (4.72 and 4.11%), and protein content (21.40 and 21.74%) was not different (*p* > 0.05) between the F- and C-samples, respectively (Table 3). However, the addition of 8% dietary fish oil affected (*p* < 0.05) fat thickness and lean meat content, detected through FOM (Fat-O-Meat) analysis within the SEUROP classification. As follows from Table 4, neither the pH values nor the color characteristics were affected by dietary fish oil; however, a strong tendency (*p* = 0.07) to decrease drip loss was established in the F-samples. On the other hand, both tested texture properties of the muscle (WBS and hardness) were substantially decreased by dietary fish oil (*p* < 0.01).

### 3.3. Deposition of Fatty Acids in the Tissues

The content of the basic groups of fatty acids in the muscle (*m. longissimus lumborum et thoracis*), liver, and heart is presented in Figure 1. Dietary fish oil decreased (*p* < 0.05) the sum of the saturated fatty acids in comparison with the control only in the liver, and had no significant effect (*p* > 0.05) on the deposition of the sum of monounsaturated fatty acids in either tested tissue, including the visceral (pelvic) adipose tissue (VAT; data not shown in Figure 1). As far as particular saturated fatty acids (SFAs) and monounsaturated fatty acids (MUFAs) are concerned, fish oil decreased (*p* < 0.05) the stearic acid (18:0; 296 vs. 431 mg/100 g of the fresh tissue) and oleic acid (18:1 n-9; 278 vs. 401 mg/100 g) content in the liver.

The sum of the polyunsaturated fatty acids n-6 was decreased (*p* < 0.05) by dietary fish oil in the liver and heart, respectively. The linoleic acid (LA; 18:2 n-6) content in the liver was decreased (*p* < 0.05) from 338 to 144 mg/100 g, and the arachidonic acid (AA; 20:4 n-6) content from 256 to 42 mg/100 g in the C- and F-pigs, respectively. A significant difference in AA content was also found in the heart, at 103 (C) and 59 (F) mg/100 g, respectively.

On the other hand, the sum of PUFAs n-3 was increased (*p* < 0.05) in all tested F-tissues (Figure 1), including VAT. The EPA (20:5 n-3) content was higher (*p* < 0.05) in the F-muscle (38 vs. 2 mg/100 g), F-liver (123 vs. 12 mg/100 g), F-heart (85 vs. 5 mg/100 g), and F-VAT (696 vs. 26 mg/100 g). The same was true (*p* < 0.05) for the DHA (22:6 n-3): 57 vs. 4 (muscle), 98 vs. 26 (liver), 55 vs. 5 (heart), and 1645 vs. 53 (VAT) mg/100 g, respectively. The PUFA n-6/n-3 ratio was lower in all F-tested tissues (*p* < 0.05): 2.1 vs. 11.0 (muscle), 0.8 vs. 7.1 (liver), 1.2 vs. 12.2 (heart), and 1.7 vs. 13.2 (VAT).

It is necessary to mention that in cases when a difference in fatty acid deposition was found between the dietary groups, these differences were not affected by the total lipid content, which did not differ between the groups in any of the tested tissues (*p* > 0.05); the values of the F- and C-groups were as follows (mean ± SEM; in %): 2.00 ± 0.09 and 2.30 ± 0.15 in the muscle, 3.13 ± 0.18 and 3.54 ± 0.16 in the liver, 1.82 ± 0.35 and 1.56 ± 0.05 in the heart, and 54.93 ± 1.96 and 49.72 ± 1.79 in the VAT, respectively.

### 3.4. Plasma Lipid Profile

No significant effect of the F-diet on plasma total cholesterol, HDL cholesterol, or LDL cholesterol was found (*p* > 0.05). However, fish oil in the tested amount increased plasma TAG levels by 260% (*p* < 0.05; Figure 2).

### 3.5. Hematological Markers

In addition to increasing (*p* < 0.05) the counts of leukocytes and neutrophil and eosinophil granulocytes in pig blood, dietary fish oil also tended to increase lymphocyte numbers (Figure 3) and significantly decreased (*p* < 0.05) hematocrit (to 35.7%) in comparison with the control (37.1%). Counts of erythrocytes (6.51 × 10^12^ L^−1^; the same value in the F- and C-groups) and the amount of hemoglobin (122.2 vs. 122.7 g L^−1^ in the F- and C-pigs) were not affected.

### 3.6. Serum Metabolites and Liver Enzyme Activities

With regard to the three enzymes used as biomarkers for liver health, dietary fish oil tended to increase (*p* = 0.10) the activity of alanine transaminase and significantly (*p* < 0.05) increased the activity of aspartate transaminase and alkaline phosphatase, respectively (Figure 4). The concentration of urea also increased (*p* < 0.05) in the F-pigs (2.83 µmol L^−1^) in comparison with the controls (2.17 µmol L^−1^). There was a tendency (*p* = 0.09) for the dietary fish oil to decrease serum creatinine (133.3 vs. 140.0 µmol L^−1^), but neither the total plasma protein (63.14 and 61.85 g L^−1^) nor the serum albumin (30.62 and 31.83 g L^−1^) level differed between the fish oil and control groups (*p* > 0.05). The same was true regarding plasma glucose levels: 6.00 and 6.22 mmol L^−1^ in the F- and C-group, respectively (*p* > 0.05).

## 4. Discussion

When considering supplementing a pig diet with PUFA n-3 based on a comparison with the results of recently published studies, it is important to take account of the source and amount of PUFA n-3 in the diet, the length of the fattening interval, and the number of animals in the experimental (control) group. Animal numbers per test group are usually in the range of eight [11] to 24 [18]. Only six pigs per group were used in the present study due to the fact that the experiment was not performed in a production facility, but in an experimental stable of the research institute with a limited capacity. However, this number is still plausible from the viewpoint of statistics.

The fattening interval usually lasts from four weeks ([11]; the present experiment) to eight weeks [12,14,19] to as long as 22 weeks [20].

As far as the quality and quantity of PUFA n-3 is concerned, in light of recent studies, fish oil was used in our previous experiment [14]. The effect of DHA-rich microalga on pig performance and pork quality was tested in the studies of [18] (0.5% of microalga) and [21] (1% of microalga); de Tonnac and Mourot [12] applied microalga in a combination with linseed (a source of α-linolenic acid, a precursor of EPA and DHA). Different ratios of linseed oil with sunflower oil (rich in n-6 linoleic acid) or linseed oil with soybean oil were used to obtain various PUFA n-6/n-3 ratios by [11], [19], and [6], respectively.

Regarding the performance traits (Table 3), according to [20], body weight gain is dependent on (among other factors) the available energy content of the diet. Because the fat (and consequently metabolizable energy) content of the F-diet was higher in the present experiment, not energy, but lower crude protein content of the F-diet (Table 1) can possibly explain the lower daily weight gain of the F-pigs in comparison with controls.

As far as the physical traits of the muscle are concerned, the absence of an effect of dietary fish oil (EPA and DHA) on muscle color characteristics established in the present study is in agreement with the data of de Tonnac and Mourot [12], who reported no effect of a microalga supplement containing 110 mg of EPA + 7030 mg of DHA per 100 g of the diet on the L*, a*, or b* values of the *longissimus dorsi* muscles of male pigs (at the age of 22 weeks). Liu and Kim [6] also found that a different PUFA n-6/n-3 ratio in a pig diet had no effect on meat (*longissimus* muscle) color. Moreover, both Liu and Kim [6] and Huang et al. [20] reported no effect of different PUFA n-6/n-3 ratios and high dietary lipid content, respectively, on the ultimate pH value or drip loss of the muscles of pigs, which is in agreement with our results (Table 4).

Contrary to other physical characteristics of the muscle that were not affected, dietary fish oil substantially improved texture acceptance by softening the meat, i.e., decreasing both the Warner-Bratzler shear force and hardness (*p* < 0.01). The reason is not clear, but it probably was not the intramuscular fat content, which was not different (*p* > 0.05) between the F- and C-group: 2.00 and 2.30 g/100 g, respectively. According to Rey et al. [22], meat from pigs receiving vitamin D_3_ in water showed reduced cohesiveness, gumminess, and chewiness. Therefore, improved meat tenderness in the present experiment could be explained by an effect of vitamin D_3_, whose content in the fish oil was 5.6 mg/kg. Contrary to our data, dietary lipid supplementation in the form of flaxseed oil (3%) and poultry fat (2%) had no effect on the Warner-Bratzler shear force of the pig *longissimus* muscle in the study of Adhikary et al. [23].

When looking at fatty acid deposition in the tissue, considering the significant above-mentioned differences between recent studies in the length of the fattening period (four to 22 weeks), Moran et al. [21] argued that supplementing with a higher concentration of desirable fatty acids for a shorter period of time can achieve similar or even better enrichment of meat than a longer-term supplementation with lower concentrations.

Recent studies that have evaluated the effect of dietary PUFA n-3 on pork enrichment have focused on the *longissimus* muscle. Liu and Kim [6] reported markedly increased levels of ALA, DPA, and DHA (1.5-, 3-, 2.5-fold) in the muscles of pigs fed a diet with a PUFA n-6/n-3 ratio of five with a corresponding reduction of LA and AA. The F-diet in the present experiment (n-6/n-3 ratio 0.9) increased (*p* < 0.05) EPA, DPA, and DHA content in the *m. longissimus*, without changing ALA, LA, or AA.

The validity of the high level of dietary fish oil supplementation (80 g/kg of feed) in the present experiment can be corroborated by the following comparison: the sum of PUFAs n-3 in the F-muscle increased more than four-fold in comparison with the control in the present experiment, while a diet with 1% and 0.5% of DHA-rich microalga increased this parameter by only 26% [21] and was not able to change it at all [18], respectively. Dietary fish oil decreased (*p* < 0.05) the PUFA n-6/n-3 ratio in the F-muscle in the present experiment from 11.0 to 2.1, while a microalga diet with either tested concentration was not able to change this ratio in comparison with the control [18,21].

The second most frequently tested pig tissue in the given context is backfat [12,18,21]. Smink et al. [11] reported that contrary to intramuscular fat, where dietary ALA increases EPA content, this is not the case in backfat. We did not test backfat in the present experiment, but rather metabolically active VAT. In this case, however, not ALA but rather high EPA and DHA content in the F-diet (Table 2) increased (*p* < 0.05) the EPA concentration in the VAT 27-fold in comparison to the control, from 26 to 696 mg/100 g (the increase was 19-fold in the muscle, from 2 to 38 mg/100 g).

Therefore, the results of the present experiment do not confirm the statement by Wood et al. [24] that PUFA n-3 are preferentially incorporated into phospholipids (PL) over triacylglycerols (TAG), which can explain their relative increase in the PL-rich muscle, as opposed to the TAG-rich backfat. On the other hand, our results are in agreement with the findings of Abbot et al. [25], that the adipose tissue TAG is the most responsive to changes in dietary PUFA content, rather than the membrane PUFA content of the tissue phospholipids, which are fairly unresponsive to greater changes in dietary PUFA content.

PUFA n-3 deposition in the main metabolic organ, the liver, is measured less often than muscle or backfat. Dietary fish oil decreased (*p* < 0.05) the content of n-6 arachidonic acid, increased (*p* < 0.05) the content of EPA and DHA, and decreased the PUFA n-6/n-3 ratio (from 7.1 to 0.8) in the liver in the present experiment, in accordance with the effect of a combination of linseed and DHA-rich microalga reported by [12]. The same results were obtained in our previous experiment in pigs with fish oil delivered in much lower amounts (2.5%) than in the present study.

When evaluating the plasma lipid profile, LC-PUFA n-3 are known [26] to decrease plasma TAG via the activation of the transcription factor PPARα (peroxisome proliferator-activated receptor α) and the inhibition of SREBP-1 (sterol response element-binding protein 1), which stimulate fatty acid β-oxidation and inhibit fatty acid synthesis. However, the F-diet conspicuously increased plasma TAG in the present experiment (Figure 2). It is plausible that it is not fatty acid composition but rather the sheer amount of fat in the F-diet (much higher than in the control diet) that is the reason for the TAG increase. Huang et al. [20] came to the same conclusion using diets with either 0% or 6% of added lipid, which resulted in a significant (*p* < 0.05) increase in the plasma TAG in pigs fed a fatty diet (0.40 vs. 0.49 mmol L^−1^). Dietary lipid supplementation also increased the serum TAG in pigs in the study of Adhikary et al. [23].

On the other hand, when using isolipidic diets, a higher dietary PUFA n-3 content is able to decrease the plasma TAG in pigs, as reported by, e.g., [6]; the plasma TAG decreased from 0.89 to 0.60 when the dietary PUFA n-6/n-3 ratio decreased from 15:1 to 5:1, which the authors explained by the ability of PUFA n-3 to inhibit hepatic triglyceride synthesis and secretion, reduce the intestinal and hepatic apolipoprotein B species that are responsible for TAG clearance, and increase lipoprotein activity. Similarly, favorable dietary n-6/n-3 ratios (1:1 or 5:1) markedly suppressed the expression levels of lipid metabolism-related genes and proteins in an experiment of Duan et al. [19].

With regard to hematological markers, fish oil at an amount of 2.5% of the diet did not change the counts of any tested white blood cells in our previous experiment [15], but tended to only increase eosinophil counts in comparison with the control palm oil diet (from 0.21 to 0.26 × 10^12^/L). However, in the present experiment, when using a much higher amount (8%) of dietary fish oil, the above-mentioned tendency manifested in a significant increase in eosinophil counts (from 0.17 to 0.39 × 10^12^/L; *p* < 0.05), as well as in the counts of neutrophils and leukocytes (Figure 3).

Added fish oil substantially increased vitamin D_3_ content in the experimental diet (by 5.6 mg/kg). Though short term supplementation of up to ten times of the recommended dose of vitamin D_3_ has no adverse effect on calcium and bone metabolism [27], the content of vitamin D_3_ was much higher in the present experiment, which could influence calcium-phosphorus metabolism.

The lack of effect of dietary fish oil on total plasma protein and plasma albumin that have been established in the present study corresponds with the data of [23]. A dietary protein adequacy in the F-diet can therefore be supposed in the present experiment, despite the lower crude protein content due to the substantial lipid supplementation (Table 1). According to Adhikary et al. [23], serum total protein or albumin levels can be a reflection of protein adequacy or positive protein metabolism. On the other hand, Huang et al. [20] reported greater serum total protein and albumin concentrations after supplementing the diet with 1% flaxseed oil and 5% poultry fat.

On the other hand, a high dietary flaxseed oil and poultry fat diet (6%) in comparison with the diet without added fat only tended to increase serum urea (from 2.50 to 2.76 µmol/L) in the study of Huang et al. [20], but the increase in the F-pigs was significant in the present experiment (2.17 vs. 2.83 µmol/L). Huang et al. [20] mentioned a possible serum urea nitrogen decrease with the increasing efficiency of feed utilization. The lower efficiency of energy utilization can be a partial explanation for the significant increase of plasma urea in F-pigs.

## 5. Conclusions

The dietary fish oil was used in the present experiment in a higher than usual amount (8%) because of unsatisfactory results with lower doses. Two hypotheses were tested: 1) the nutritive value of pork (content of LC-PUFA n-3; PUFA n-6/n-3 ratio) will be substantially increased; 2) selected physiological markers, including plasma lipid levels, will be improved in a pig used as an animal model of human nutrition. A contradictory effect of the tested dietary level of fish oil can be concluded from this viewpoint.

EPA and DHA content was significantly increased and the PUFA n-6/n-3 ratio substantially decreased in the F-variants of all tested pig tissues; moreover, one of the tested meat texture characteristics was significantly improved (decreased hardness). On the other hand, the second hypothesis was not confirmed: dietary fish oil increased plasma TAG level, counts of leukocytes and neutrophils, and activities of two (of the three tested) liver enzymes.

Therefore, consumption of pork substantially enriched with LC-PUFA n-3 can be fully recommended, but direct consumption of fish oil by humans, though also recommendable, should be considered with caution.

## Figures and Tables

**Figure 1 animals-10-01657-f001:**
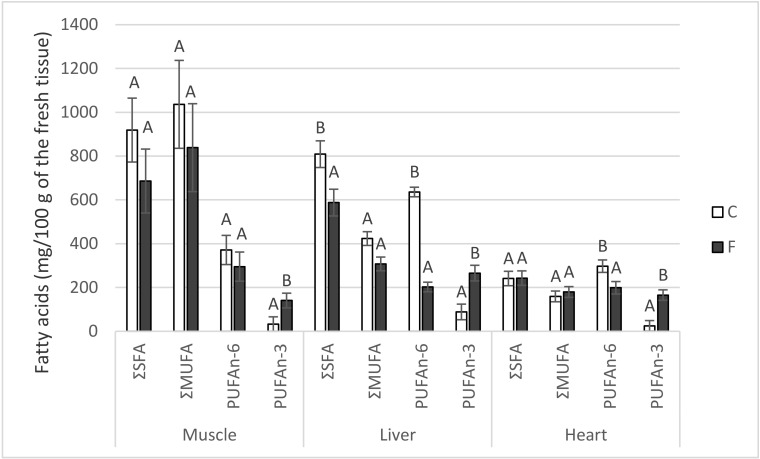
Deposition of fatty acids in the selected tissues of pigs fed for four weeks on either the control standard feed mixture (C) or feed mixture enriched with 8% fish oil (F); SFA—saturated fatty acids (myristic, palmitic, heptadecanoic, stearic); MUFA—monounsaturated fatty acids (palmitoleic, oleic); PUFA—polyunsaturated fatty acids (n-6: linoleic, γ-linolenic, eicosadienoic, homo- γ-linolenic, arachidonic, docosatetraenoic; n-3: α-linolenic, eicosapentaenoic, docosapentaenoic, docosahexaenoic). Means with different letters within the given parameter differ at *p* < 0.05 (independent groups *t*-test; *n* = 6).

**Figure 2 animals-10-01657-f002:**
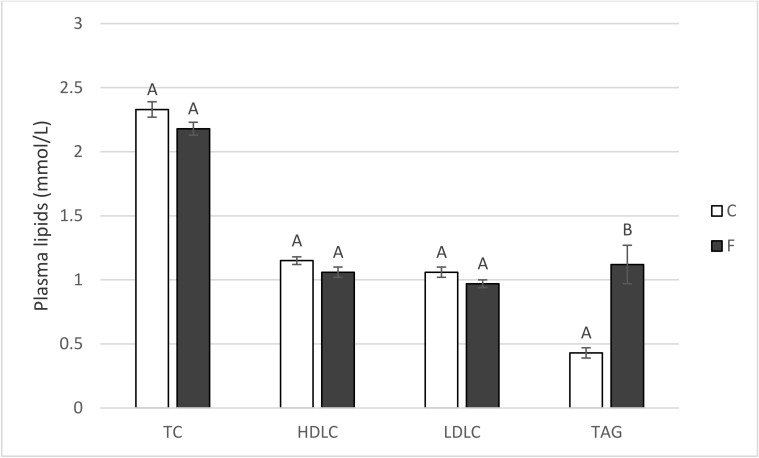
Concentration of total cholesterol (TC), high-density lipoprotein cholesterol (HDLC), low-density lipoprotein cholesterol (LDLC) and triacylglycerols (TAG) in the plasma of pigs fed for four weeks on either the control standard feed mixture (C) or feed mixture enriched with 8% fish oil (F). Means with different letters within the given parameter differ at *p* < 0.05 (independent groups t-test; *n* = 6).

**Figure 3 animals-10-01657-f003:**
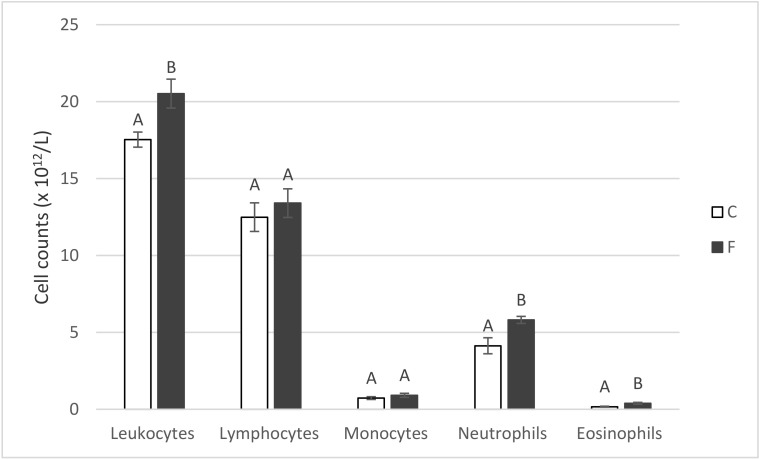
Selected hematological parameters of pigs fed for four weeks on either the control standard feed mixture (C) or feed mixture enriched with 8% fish oil (F). Means with different letters within the given parameter differ at *p* < 0.05 (Mann–Whitney U test; *n* = 6).

**Figure 4 animals-10-01657-f004:**
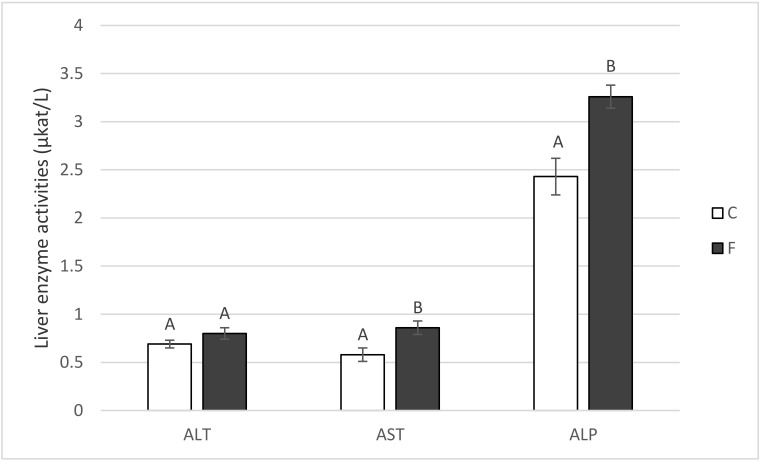
Activities of the liver enzymes alanine transaminase (ALT), aspartate transaminase (AST) and alkaline phosphatase (ALP) of pigs fed for four weeks on either the control standard feed mixture (C) or feed mixture enriched with 8% fish oil (F). Means with different letters within the given parameter differ at *p* < 0.05 (Mann–Whitney U test; *n* = 6).

**Table 1 animals-10-01657-t001:** Composition of the diets.

Composition (g/kg of the Feed Mixture; Dry Matter Content 88%)		Diet	
		C ^1^	F ^2^
Components (g/kg)			
	Basic feed mixture ^3^	1000	920
	Fish oil	/	80
Nutrients (g/kg)			
	Crude protein ^4^	144	133
	Lysine ^5^	8.2	7.5
	Methionine ^5^	2.6	2.4
	Fat ^6^	36	108
	Crude fiber ^7^	65	60
	Ash	48	44
	Calcium ^5^	6.0	5.5
	Phosphorus ^5^	4.3	4.0
	Sodium ^5^	1.6	1.5
	Nitrogen-free extractives ^8^	707	655
ME ^9^ (MJ/kg)		13.8	15.5

^1^ Control diet; ^2^ diet enriched with 8% of fish oil; ^3^ basic pelletized complete feed mixture for pig fattening (De Heus, Marefy, Czech Republic; formulated according to the “Nutrient Requirements of Swine” National Research Council [16]), composition (the producer refused to communicate percentages of particular components due to the trade secret): wheat, barley, wheat bran, wheat middlings, dark distillery stillages, rapeseed expellers, vinas, sodium carbonate, animal fat (pork lard), salt, premix of vitamins, and minerals: vitamin A 4500 IU/kg, vitamin D_3_ 1100 IU/kg, vitamin E 54 mg/kg, MnO_2_ 25 mg/kg, FeSO_4_ · 7H_2_O 50 mg/kg, Na_2_SeO_3_ 0.2 mg/kg, ZnO 60 mg/kg; ^4^ determined using KD-310-A-1015 KjelROC Analyzer (Furulund, Sweden); ^5^ declared by the producer; ^6^ hexane/2-propanol extract; ^7^ determined using ANCOM220 Fiber Analyzer (Ancom Technology, Macedon, NY, USA); ^8^ calculated as a remainder to 100%; ^9^ calculated from nutrient content.

**Table 2 animals-10-01657-t002:** Fatty acid content in fish oil and in the diets (sixteen fatty acids identified according to the retention times of the standard are presented).

Fatty Acid	Fatty Acid Content		
	Fish Oil (%) ^1^	Diets (mg/100 g of the Feed Mixture)	
		C ^2^	F ^3^
14:0	4.6	18.9	112.6
16:0	11.1	347.3	498.8
17:0	1.0	5.7	16.8
18:0	2.9	135.0	139.7
16:1	10.4	38.7	265.3
18:1	19.5	570.0	848.0
20:1	15.8	34.6	407.5
18:2n-6	3.3	501.1	546.0
18:3n-6	0.8	0.2	0.2
20:2n-6	1.0	5.3	14.8
20:4n-6	1.0	5.4	15.0
22:4n-6	1.1	0.3	14.8
18:3n-3	1.6	38.3	62.4
20:5n-3	10.4	14.8	247.1
22:5n-3	1.9	7.9	36.5
22:6n-3	13.6	22.3	328.9
n-6/n-3	0.18	6.15	0.87

^1^ Of the sum of total determined fatty acids; ^2^ control—standard commercial feed mixture; ^3^ standard feed mixture enriched with 8% (*w*/*w*) of fish oil (commercial *jecoris aselli oleum*).

**Table 3 animals-10-01657-t003:** Performance characteristics of pigs fed either a standard diet (C) or a diet enriched with 8% fish oil (F).

Trait	Sample (Mean ± SEM)						*p*-Value ^1^
	C			F			
Final live weight (kg)	79.17	±	2.55	75.58	±	1.76	0.28
Daily weight gain (kg/day)	0.92	±	0.05	0.75	±	0.02	0.01
Weight at delivery (kg)	60.79	±	1.77	57.85	±	1.44	0.23
Yield (kg)	48.35	±	2.93	43.85	±	2.05	0.24
Shoulder weight (kg)	4.89	±	0.16	4.84	±	0.18	0.85
Leg weight (kg)	9.27	±	0.29	8.61	±	0.21	0.10
Fat thickness (mm) ^2^	8.67	±	0.33	10.17	±	0.48	0.02
Muscle thickness (mm) ^2^	52.67	±	1.20	52.33	±	1.52	0.87
Lean meat (%) ^2^	63.88	±	0.25	63.03	±	0.19	0.02
Dry matter (%) ^3^	26.38	±	0.38	26.86	±	0.79	0.59
Fat (%) ^3^	4.11	±	0.55	4.72	±	0.99	0.61
Protein (%) ^3^	21.74	±	0.36	21.40	±	0.30	0.48

^1^ Mann–Whitney U test (*n* = 6); ^2^ SEUROP pig carcass classification with FOM (Fat-O-Meat) method; ^3^
*musculus longissimus lumborum et thoracis* analysis.

**Table 4 animals-10-01657-t004:** Physical characteristics of the *longissimus dorsi* muscle of pigs fed either a standard diet (C) or a diet with 8% fish oil (F).

Trait	Sample (Mean ± SEM)						*p*-Value ^1^
	C			F			
pH_1_ ^2^	6.52	±	0.08	6.71	±	0.11	0.21
pH_24_ ^3^	5.53	±	0.02	5.56	±	0.01	0.25
Drip loss (%)	6.87	±	0.48	5.46	±	0.51	0.07
L* ^4^	55.78	±	1.20	55.64	±	1.36	0.94
a* ^5^	1.58	±	0.37	1.59	±	0.73	0.99
b* ^6^	10.86	±	0.51	11.71	±	0.64	0.32
WBS ^7^ (N)	48.81	±	0.87	40.19	±	0.67	0.00
Hardness (N)	173.0	±	7.57	135.1	±	2.91	0.00

^1^ Mann–Whitney U test; *n* = 6; ^2^ one hour after the slaughter; ^3^ 24 h after the slaughter; ^4^ lightness from black to white; ^5^ lightness from green to red; ^6^ lightness from blue to yellow; ^7^ Warner-Bratzler shear force.

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
