# Peer review of "Effect of High Dietary Level (8%) of Fish Oil on Long-Chain Polyunsaturated Fatty Acid n-3 Content in Pig Tissues and Plasma Biochemical Parameters"

_animals, 2020, doi:10.3390/ani10091657_

Round 1
Reviewer 1 Report
The revisions clearly addressed the concerns/comments of the reviewers
Author Response
Dear Reviewer,
thank you for your Review report aiming to improve our manuscript.
Tomas Komprda, Miroslav Jůzl
Reviewer 2 Report
Manuscript ID: animals-937515
Article Title: Effect of high dietary level (8 %) of fish oil on long-chain polyunsaturated fatty acid n-3 content in pig tissues and plasma biochemical parameters.
I thank the Authors for their answers, however, more questions arise here:
Line 117, page 3, Table 1: According to which nutritional requirements for pigs are the feed mixtures formulated?
Page 3, Table 1: Does the chemical composition of the mixtures in Table 1 refer to 1 kg of DM? Please specify this in the table.
Page 4: Table 2 still shows incorrect amounts of individual fatty acids in diets. The sum of fatty acids (C diet = 17.46 g / 1 kg and F diet = 35.54 g/1 kg) does not correspond to the level of fat in the mixtures in table 1 (C diet = 36 g / 1 kg and F diet = 108 g/1 kg).
In Table 2, the content of the individual fatty acids refers to a 1 kg of mixture or 1 kg of dry mass of the mixture?
Coming back to the problem with vitamin D. It became an additional factor in the study, but its influence was not included in the statistical analysis. And in the case of the F diet, the addition of vitamin D in the premix is unnecessary.
Author Response
Thank you very much for your comments aiming to improve our manuscript.
Line 117, page 3, Table 1: According to which nutritional requirements for pigs are the feed mixtures formulated?
Answer: Feed mixtures (including the feed mixture that was used in the present experiment) are currently formulated according to the “Nutrient Requirements of Swine”, 11th revised edition, by National Research Council, 2012. We used a commercial feed mixture produced by the firm De Heus, Marefy, Czech Republic (see Table 1 of the manuscript) that is obliged to abide by the Requirements. We also added this information to Table 1 of the revised manuscript.
Page 3, Table 1: Does the chemical composition of the mixtures in Table 1 refer to 1 kg of DM? Please specify this in the table.
Answer: The chemical composition of the mixtures refer not to 1 kg of dry matter, but to 1 kg of the mixture, with a further specification of dry matter content that was 88 %. We specified it in Table 1.
Page 4: Table 2 still shows incorrect amounts of individual fatty acids in diets. The sum of fatty acids (C diet = 17.46 g / 1 kg and F diet = 35.54 g/1 kg) does not correspond to the level of fat in the mixtures in table 1 (C diet = 36 g / 1 kg and F diet = 108 g/1 kg).
Answer: The amounts of individual fatty acids in Table 2 are correct. There are two reasons why the sum of fatty acids does not equal to the level of fat: 1) Based on the retention times of the standard we used in the analysis, sixteen peaks were identified, but altogether 44 peaks were present in the chromatograms; it is necessary to note, however, that the applied standard of fatty acid methyl esters covered all physiologically (nutritionally) important fatty acids. 2) Even if all chromatogram peaks could be identified (which would be useless), the sum would still be much lower than the amount of fat, because apart from fatty acids, fat (lipids) contain also great amounts of cholesterol, glycerol, free fatty acids (that are not identifiable by the given method), etc.
In Table 2, the content of the individual fatty acids refers to a 1 kg of mixture or 1 kg of dry mass of the mixture?
Answer: The content of the individual fatty acids in Table 2 refers to 100 g of the feed mixture; we added this information to Table 2.
Coming back to the problem with vitamin D. It became an additional factor in the study, but its influence was not included in the statistical analysis. And in the case of the F diet, the addition of vitamin D in the premix is unnecessary.
Answer: We knew vitamin D content in the diets based on the data of the producer of the feed mixture and the producer of fish oil. However, despite the fact that vitamin D content can be added as an independent variable into the statistical model, we neither measured its content in the pigs’ tissues nor assessed its effects on the pigs’ metabolism. Therefore we were not able to include vitamin D in the statistical analysis. Based on your previous comment regarding this topic (thank you very much for the reminder), we realized that vitamin D content in the experimental diet was much higher than in the control one. However, based on the literature data (that we quoted in the R1 version of the manuscript), an effect of vitamin D manifests itself after several months of supplementation. Therefore, we suppose no substantial effect of vitamin D on the pigs’ metabolism in our study lasting only three weeks. We fully agree with you that the addition of vitamin D in the premix of the F diet was unnecessary. However, we obtained from the producer the complete feed mixture, with the premix as an integral part of it, and it was impossible to separate vitamin D off the mixture. Even if it would have been possible, the vitamin D content in the F-diet would decreased only by 0.5 %.
This manuscript is a resubmission of an earlier submission. The following is a list of the peer review reports and author responses from that submission.
Round 1
Reviewer 1 Report
Komprda et al. fed pigs diet supplemented with a high dose of fish oil and explored the effect of the fish oil on the meat composition (as a potential source of omega-3 PUFA for humans) and on metabolic parameters of the pigs.
The manuscript is complicated by the fact that two objectives are mixed together. When animals (pigs) are used as a source of meat, the fattening and growth is expected, and this is not metabolically beneficial scenario. They show that with a high dose of a fish oil, omega-3 PUFA in the meat are elevated. The intervention also (negatively) affected several biochemical parameters of the pigs, but the relevance or rationale of such observation is missing. Overall, the scientific hypothesis is missing and the manuscript describes a feeding experiment with the expectable output.
If I consider that the novelty is a way how to increase omega-3 PUFA content in pork meat, several questions have to be answered.
"jecoris aselli oleum" is most probably a cod liver oil. Apart from fatty acid composition, no characteristics of the oil are shown.
Would it be economical to feed pigs with such oil product? What is the rationale for feeding pigs with high doses of cod liver oil? How pigs metabolize/digest the cod liver oil? From the plasma TAG levels and liver enzyme markers, these animals were "less healthy" than the control animals on "fattening" diet. Thus, is such a meat of better quality?
Composition of the diet is incomplete. E.g. carbohydrate content is missing. Food consumption data are missing.
I do not understand the rationale for the white blood cells analysis. How is this relevant?
Discussion is very long and not focused. It should be significantly shortened.
The manuscript could be shortened and focused only on one story.
Author Response
The manuscript is complicated by the fact that two objectives are mixed together. When animals (pigs) are used as a source of meat, the fattening and growth is expected, and this is not metabolically beneficial scenario. They show that with a high dose of a fish oil, omega-3 PUFA in the meat are elevated. The intervention also (negatively) affected several biochemical parameters of the pigs, but the relevance or rationale of such observation is missing. Overall, the scientific hypothesis is missing and the manuscript describes a feeding experiment with the expectable output.
- Answer: Thank you for your opinion. Our study had two hypotheses, we described in the introduction and in the conclusion (marked in the revised manuscript – lines 78-82 and 438-442). First – to evaluate quality of pork, physical properties and composition of FA, especially content of LC-PUFA n-3 by dietary intervention. Second – we expected improvement of selected plasma lipid metabolism markers (triglicerides, total and LDL cholesterol) by fish oil enriched diet. PUFA n-3 and PUFA n-6/n-3 ratio were significantly increased and decreased, respectively. Therefore, this hypothesis was confirmed. On the other hand, the second hypothesis was not confirmed: dietary fish oil increased plasma TAG level, counts of leukocytes and neutrophils and activities of two (of the three tested) liver enzymes if compared with standard diet.
If I consider that the novelty is a way how to increase omega-3 PUFA content in pork meat, several questions have to be answered.
"jecoris aselli oleum" is most probably a cod liver oil. Apart from fatty acid composition, no characteristics of the oil are shown.
- Answer: Jecoris aselli oleum - is a purified oil obtained from fresh liver of cod (Gadus morhua ) and other species of the family Gadidae, solids are removed by cooling and filtration. It is analysed according to the requirements of Czech and the European Pharmacopoeia. Fish oil is defined by first column in Table 2. and this analysed content corresponds to the requirements of the Czech and European Pharmacopoeia.
Would it be economical to feed pigs with such oil product? What is the rationale for feeding pigs with high doses of cod liver oil? How pigs metabolize/digest the cod liver oil? From the plasma TAG levels and liver enzyme markers, these animals were "less healthy" than the control animals on "fattening" diet. Thus, is such a meat of better quality?
- Answer: Economic aspects of this experiment were not evaluated, but the high prize of such product could be expected. Yes, we used medicinal fish oil, because it´s composition is defined. But, the feed industry could obtain cheaper variants.
- Plasma parameters of lipid metabolism improvement was expected in the group with fish oil enriched diet as well as antiinflamatory effect of improoved n-6/n-3 PUFA ratio. Surprisingly, we found deteriorated plasma lipids, in contrary with other authors, who fed animals with more healthy n-6/n-3 ratio containing diet. From the point of view of triglicerides in plasma, animals with fish oil diet are less healthy, but their meat has favourable n-6/n-3 ratio and therefore is more suitable for human nutrition than “conventional” pork. Deteriorateted lipid metabolism parameters of experimental animals could be considered as a model of possible limitation in using fish oil for direct human consumtion as a part of healthy diet.
Composition of the diet is incomplete. E.g. carbohydrate content is missing. Food consumption data are missing.
- Answer: In lines 99-101: The animals had free access to drinking water and were fed twice a day ad libitum. By subtracting leftovers, the net feed consumption and intake was measured per pen (not individually). We added text in the line 95: each group divided into two pens. Carbohydrate content could be counted from the total energy content by subtracting energy obtained from crude protein and fat (data in Table 1).
I do not understand the rationale for the white blood cells analysis. How is this relevant?
- Answer: Long chain PUFA n-3 and n-6 affect the immune system function (line 407-409). In our previous experiments with 2.5% fish oil addition, we did not find any changes. We wanted to determine the response of the immune system to the altered long chain PUFA ratio.
Discussion is very long and not focused. It should be significantly shortened. The manuscript could be shortened and focused only on one story.
- Answer: In our opinion, discussion chapter is dedicated for whole aspects other studies, but we have removed some parts, which are not directly connected to our results.

Reviewer 2 Report
Dear Authors:
A very nice straight forward study on PUFA (here fish oil) on n3 PUFA content of pork. Sometimes pigs lipid metabolism is not quite as easy to compare to rodents and humans. In the first and most important instance pigs do not exhibit liver de novo fatty acid synthesis. In fact PUFA have really only been well studied to suppress liver lipogenesis in rodents. But the subject of this MS is the role f 8% fish oil on n-3 PUFA content in pigs.
There is no doubt that one can do this as shown well here; but 8% fish oil would have its own issues on oxidative state as well as organoleptic properties in pork. North American consumers would likely reject this type of pork irrespective of how "healthy it was". Whenever I go to Europe and bite into fishmeal fed chicken I stop eating it.
Yet this study was well designed and carried out and the introduction/discussion will be useful to workers in the field.
Reference 19 needs a correction:
Bergen, WG is a coauthor of this paper
Author Response
- Answer: Thank you, we corrected all author´s names.
- (highlighted in manuscript by pink colour)
- Huang, C.; Chiba, L.I.; Magee, W.E.; Wang, Y.; Griffing, D.A.; Torres, I.M.; Rodning, S.P.; Bratcher, C.L. Bergen, W.G.; Spangler, E.A. Effect of flaxseed oil, animal fat, and vitamin E supplementation on growth performance, serum metabolites, and carcass characteristics of finisher pigs, and physical characteristics of pork. Livest Sci 2019, 220, 143-151. doi: 10.1016/j.livsci.2018.11.011.

Reviewer 3 Report
Manuscript ID: animals-904651
Title: Effect of high dietary level (8 %) of fish oil on long-chain polyunsaturated fatty acid n-3 content in pig tissues and plasma biochemical parameters
The impact of dietary fat sources on pig meat quality, fatty acids composition and also on the sensory attributes of pork is the subject of many studies. We are observing a continuous increase in consumer interest in maintaining and improving health by eating functional foods. In general, so the topic of the article is current and interesting, but the manuscript needs to be improved.
Detailed comments:
Simple Summary:
Line 18-20 page 1: Such a conclusion is an over-interpretation of the results of the research carried out.
Materials and Methods
Line 98 page 3: Based on what nutritional requirements pig diets have been optimized?
Page 3: There is no detailed composition of the feed mixture in Table 1. What was the main source of fat in the control diet? What feed fat was used?
Page 4: Table 2 shows incorrect amounts of individual fatty acids in diets.
Line 103 page 3: Was blood collected from animals after fasting? Perhaps the access of animals to feed contributed to the high level of TG in the blood of pigs in the group fed with the fish fat mixture.
Statistical analysis
Line 183 page 5: Were the data not normally distributed? The non-parametric test (Mann-Whitney U test) was chosen for some analyses. Parametric tests are characterized by greater power and better interpretability of the obtained results. Why, has no attempt been made to transform the data so as to obtain a normal distribution?
Discussion
Why did the discussion not take into account the effect of a high dose of dietary vitamin D with fish oil?
There are many references in the literature to the influence of vitamin D on the calcium-phosphorus metabolism, on the characteristics of bone tissue and on the immune system of pigs.
Research was also conducted on obtaining meat with increased tenderness through increased supplementation of vitamin D3.
Author Response
The impact of dietary fat sources on pig meat quality, fatty acids composition and also on the sensory attributes of pork is the subject of many studies. We are observing a continuous increase in consumer interest in maintaining and improving health by eating functional foods. In general, so the topic of the article is current and interesting, but the manuscript needs to be improved.
- Answer: Thank you for the review and we will revise our manuscript according to your recommendations.
Detailed comments:
Simple Summary:
Line 18-20 page 1: Such a conclusion is an over-interpretation of the results of the research carried out.
- Answer: We revised our last sentence of simple summary: „…but a direct human consumption of fish oil should be limited and in accordance with recommendation of moderate intake“. Also we revised this statement in the conclusion (line 447).
Materials and Methods
Line 98 page 3: Based on what nutritional requirements pig diets have been optimized?
- Answer: Fatty acid composition in pork fat is not optimal from the point of view of human nutrition. Based on our previous research, we did not reach desirable changes in FA composition with addition of 2.5% fish oil. Our goal was to increase n-3 PUFA in tissues by fish oil suplementation.
Page 3: There is no detailed composition of the feed mixture in Table 1. What was the main source of fat in the control diet? What feed fat was used?
- Answer: The declared composition is given in the note to Table 1. (De Heus, Marefy, Czech Republic). Animal fat is commercial lard used for feed mixture from slauterhause.
Page 4: Table 2 shows incorrect amounts of individual fatty acids in diets.
- Answer: We apologise for the typos in listed units, corrected units for fatty acid content in the Table 2 are mg/100 g.
Line 103 page 3: Was blood collected from animals after fasting? Perhaps the access of animals to feed contributed to the high level of TG in the blood of pigs in the group fed with the fish fat mixture.
- Answer: Yes, blood samples was drawn from animals after fasting. Animals were fed twice a day with fresh food, blood was drawn before first feeding.
Statistical analysis
Line 183 page 5: Were the data not normally distributed? The non-parametric test (Mann-Whitney U test) was chosen for some analyses. Parametric tests are characterized by greater power and better interpretability of the obtained results. Why, has no attempt been made to transform the data so as to obtain a normal distribution?
- Answer: Yes, parametric tests are characterized by greater power and better interpretability of the obtained results. However, due to the smaller numbers of n, we proceeded to use a non-parametric test (Mann-Whitney U test).
Discussion
Why did the discussion not take into account the effect of a high dose of dietary vitamin D with fish oil?
Answer: Viamin D effect was not followed in this study. We agree that there is much evidence on modulation of immune system by vitamin D and its positive effect in resistancy to infection and protection against autoimmunity in human research as well as in animal experimental studies. Positive effect on human immunity boosting was observed in long term studies (> 4 - 6 months), whereas our experiment lasted 30 days. Vitamin D intake in our experiment was approximately 18,000 IU/animal/day (as calculated from data of fish oil producer). Existing animal studies with high doses of vitamin D supplementation that lasted similar period are not consistent with our data. Decrease in specific lymphocyte subclasses was found in pigs fed with high doses of vitamin D fortified diet by von Rosenberg et al. (2016), whereas we have found increased leukocyte count (lymphocyte fraction and subclasses were not determined). We agree, that white cell count can be affected by high dose supplementation of vitamin D, but we cannot prove it in our study.
There are many references in the literature to the influence of vitamin D on the calcium-phosphorus metabolism, on the characteristics of bone tissue and on the immune system of pigs.
Answer: We agree, that the major role of vitamin D concerns calcium and phosphate metabolism and bone mineralization. It follows from literature that short term supplementation of up to 10 times over dose of vitamin D has no adverse effect on calcium and bone metabolism (von Rosenberg et al., 2016).
Research was also conducted on obtaining meat with increased tenderness through increased supplementation of vitamin D3.
- Answer: As concerning vitamin D effect on meat quality, there was found improvement in some parameters in some studies. E.g. recently, the effect of high doses vitamin D in drinking water (500,000 IU/animal -1.200,000 IU/animal) on improvement of pork quality was observed (Ray et al., 2020).
(highlighted in manuscript by red colour)
